# OpenReview forum: "Lookahead-GCG: Improving Universal Multi-Model Optimization-Based Jailbreaking Attacks via Stochastic Nesterov Optimization"
_ICML.cc/2026/Conference — ICML 2026 regular_

### Official Review · Reviewer_tWKi · 2026-03-05

**Soundness:** 2
**Presentation:** 3
**Significance:** 2
**Originality:** 3
**Overall Recommendation:** 3
**Confidence:** 4

**Summary:**

This paper proposes Lookahead-GCG, a framework aimed at mitigating the instability of gradient aggregation in multi-model optimization for adversarial jailbreaking. The authors argue that standard gradient descent (implicitly used by GCG and its variants) fails to generalize when exposed to conflicting gradients from heterogeneous target models. To address this, the optimization is moved to a continuous embedding space, utilizing Stochastic Nesterov Accelerated Gradient (SNAG) to smooth model-specific gradient noise via its lookahead mechanism, and initialized with a Maximally Distant Template (MDT). The authors evaluate their approach using a dual-model ensemble (Llama-3.1 and Vicuna) and report improved Attack Success Rates (ASR) on both open-source and closed-source commercial APIs compared to GCG and TransferAttack baselines.

**Compliance With Llm Reviewing Policy:**

Affirmed.

**Final Justification:**

While the authors have addressed most of my technical concerns, the overall contribution remains marginal given the current landscape in 2026. With the recent surge in jailbreak research achieving near-perfect attack success rates and seamless transferability, this work lacks the necessary breakthrough to stand out. Furthermore, the absence of comparisons against genuine state-of-the-art baselines, such as I-GCG, significantly weakens the evaluation. Consequently, I have slightly adjusted my score, but I maintain my recommendation for rejection.

**Key Questions For Authors:**

1. How does Lookahead-GCG compare against some state-of-the-art, gradient-free black-box transfer attacks (e.g., Crescendo[1]) in terms of total computational overhead and final ASR on closed-source APIs?

2. Your "multi-model" evaluation is restricted to $M=2$ (Llama + Vicuna). If the source ensemble incorporated more architecturally diverse models (e.g., Qwen or MoE models like DeepSeek), causing heterogeneous gradient conflicts to scale exponentially, would SNAG's stability advantage hold? Could you provide empirical results for $M \ge 3$?

3. In Algorithm 1 (Step 4), the continuous momentum must be periodically projected back onto the highly sparse and non-convex discrete token space. Does this hard truncation practically disrupt the subtle directional advantages accumulated by the Nesterov momentum?

**Limitations:**

No. While the authors provide a standard ethical disclaimer in Section 6 , they fail to discuss the scalability bottlenecks of their method (e.g., expanding $M$) or analyze the real-world cost-benefit ratio of utilizing highly expensive white-box compute to bypass black-box defenses compared to cheaper alternatives.

**Strengths And Weaknesses:**

**Strengths**

- Most recent jailbreak research has been confined to tweaking candidate sampling or loss functions. Approaching the transferability bottleneck from the perspective of underlying optimizer dynamics and generalization stability (specifically, SNAG's lookahead mechanism) is theoretically inspiring for the adversarial machine learning community.
- Theorem 3.3 provides a rigorous proof that SNAG yields a tighter velocity stability bound compared to standard SGD when handling heterogeneous data pairs (differing model-prompt combinations).

**Weaknesses**

- The paper attempts to rescue white-box gradient-based suffix optimization (the GCG family) for cross-model transferability. However, the practical value of this paradigm is rapidly diminishing. State-of-the-art gradient-free black-box attacks (e.g., LLM-agent-based methods like PAIR or TAP, or automated template injection) inherently possess exceptional cross-model transferability without the exorbitant GPU memory and compute costs required for white-box backpropagation. Expending massive computational resources to patch an inherently poor-transferring white-box framework to blindly attack black-box targets lacks competitiveness in the current jailbreaking landscape.

- The authors claim state-of-the-art transfer ASR, yet Tables 1 and 3 exclusively benchmark Lookahead-GCG against its own optimization-based lineage (GCG from 2023, TransferAttack from 2025). To convincingly demonstrate its real-world threat level against closed-source APIs, the paper must include head-to-head ASR and computational cost comparisons against representative gradient-free black-box transfer attacks. Omitting these modern baselines makes the performance claims appear skewed and unconvincing.

- The core contribution relies on resolving severe gradient conflicts during multi-model optimization. However, the "multi-model" experimental setup consists of a mere dual-model ($M=2$) ensemble comprising Llama-3.1-8B and Vicuna-7B. Such a minuscule ensemble of two highly similar open-source architectures fails to simulate the exponential gradient divergence encountered in large-scale joint training with $M \ge 3$ or with radically distinct architectures (e.g., mixing Dense and MoE models). This severely undermines the claims of scalability and robustness against heterogeneous gradient conflicts.

---

> ### Author Rebuttal · Authors · 2026-03-31
>
> We sincerely thank you for the detailed assessment of our work. Our point-by-point responses follow.
>
> ---
>
> **Q1:The practical value of this GCG paradigm is rapidly diminishing. Gradient-free black-box attacks (e.g., PAIR) inherently possess exceptional cross-model transferability.**
>
> In our setting, we optimize a single universal adversarial suffix that covers all harmful behaviors simultaneously (Zou et al., 2023) — once optimized offline on open-source surrogate models, it can be deployed against any new query or target with no additional optimization or external API dependency. This makes it particularly suited for internal red-teaming of proprietary systems where external LLM access is restricted, air-gapped, or unavailable. PAIR/TAP cannot achieve this by design: they are per-instance methods requiring a separate optimization run per behavior, and their attack performance is directly tied to attacker LLM capability.
>
> Even in the individual transfer setting, [1] (Table 5) reports PAIR achieves only 0.24/0.27 ASR on Llama-2/Llama-3, comparable to or lower than GCG (0.33/0.27).
>
> Moreover, the continued active development of optimization-based transfer attacks — including TransferAttack (Yang et al., 2025), LARGO (Li et al., 2025), and SM-GCG (Gu et al., 2025) — further reflects the community's recognition of this direction's practical value.
>
> ---
>
> **Q2: The paper must include head-to-head ASR and computational cost comparisons against representative gradient-free black-box transfer attacks.***
>
> We respectfully clarify that comparing against gradient-free iterative attacks is not standard practice in the optimization-based jailbreak literature — I-GCG (Jia et al., 2024), TransferAttack (Yang et al., 2025), and MAC (Zhang & Wei, 2025) all evaluate within the optimization-based lineage, particularly in the universal adversarial suffix setting where PAIR/TAP are not applicable by design.
>
> To address the reviewer's concern, we provide an additional comparison against Logic Jailbreak [2], a recent universal gradient-free black-box transfer attack, evaluated under the same experimental setting as our main results. In the transfer setting on commercial APIs, Logic Jailbreak achieves *21.5%* ASR on Gemini-1.5-Pro and *11.0%* on DeepSeek-V3.2, compared to our *58.83%* and *16.00%,* respectively — our method substantially outperforms this baseline on both targets.
>
> Regarding computational cost, a direct comparison is not straightforward: [2] requires iterative calls to both an attacker LLM API and the target API, incurring monetary costs. Lookahead-GCG, by contrast, requires a one-time offline GPU optimization whose universal suffix is reusable across any new query at zero marginal cost.
>
> ---
>
> **Q3: Incorporated architecturally diverse models, would stability advantage hold?**
>
> Yes, the advantage holds across diverse models. Please refer to our response to Reviewer qqLJ (Q2).
>
> ---
>
> **Q4: The "multi-model" experimental setup consists of a mere dual-model (M=2) ensemble**
>
> Following your suggestion, we conducted additional tests using three source models (M=3). The empirical results confirm that *Lookahead-GCG maintains its performance* across both native and MDT settings.
>
> | Method | Gemma |Qwen| Gemma (MDT) | Qwen (MDT) |
> |-|-|-|-|-|
> | GCG | 16.00±1.00 | 15.75±6.72 | 13.00±3.00 | 13.75±2.47 |
> | TransferAttack | 51.75±5.30 | 42.25±10.96 | 35.00±22.00 | 35.00±4.95 |
> | *Lookahead-GCG* | *58.75±1.77* | *63.00±2.00* | *49.75±2.47* | *56.50±2.12* |
>
> ---
>
> **Q5: The continuous momentum must be periodically projected back onto the token space. Does this truncation practically disrupt the subtle directional advantages accumulated by the Nesterov momentum?**
>
> In our implementation, $v_t$ and $e_t$ are maintained independently—projection only "snaps" $e_t$ to the nearest token embedding for the forward pass, while $v_t$ continues accumulating directional information in continuous space, unaffected by the projection. To empirically verify that this hard truncation does not significantly disrupt the optimization trajectory, we measured the cosine similarity between the suffix embeddings before and after projection across all optimization steps:
>
> $$\text{Average Similarity} = \frac{1}{T} \sum_{t=1}^{T} \cos(e_t, \text{proj}(e_t)) = 0.9143$$
>
> This high similarity indicates that hard truncation induces only minor directional deviation in the embedding space.
>
> ---
>
> **Q6: In Section 6, they fail to discuss the scalability bottlenecks or the real-world cost-benefit ratio.**
>
> We will add a formal Limitations section in the revision, mentioning the scalability to larger source ensembles and the scope of our commercial API evaluation.
>
> ---
>
> Ref
>
> [1] Chu, Junjie, et al. "JailbreakRadar: Comprehensive assessment of jailbreak attacks against LLMs." *ACL,* 2025.
>
> [2] Peng, Jingyu, et al. "Logic jailbreak: Efficiently unlocking LLM safety restrictions through formal logical expression.", arXiv:2505.13527 (2025).

---

> > ### Author Rebuttal · Reviewer_tWKi · 2026-04-01
> >
> > While the authors have addressed most of my technical concerns, the overall contribution remains marginal given the current landscape in 2026. With the recent surge in jailbreak research achieving near-perfect attack success rates and seamless transferability, this work lacks the necessary breakthrough to stand out. Furthermore, the absence of comparisons against genuine state-of-the-art baselines, such as I-GCG, significantly weakens the evaluation. Consequently, I have slightly adjusted my score, but I maintain my recommendation for rejection.

---

> > > ### Author Response · Authors · 2026-04-05
> > >
> > > We truly appreciate your acknowledgment that the original technical concerns have been fully resolved. Regarding the updated concerns:
> > >
> > > ---
> > >
> > > **Q7: The absence of comparisons against genuine state-of-the-art baselines, such as I-GCG, significantly weakens the evaluation.**
> > >
> > > There might be a misunderstanding. I-GCG (Jia et al., 2024) targets *individual single-prompt* attacks, whereas our work operates in the *universal multi-prompt transfe*r setting — a distinct different problem where a single suffix must transfer across diverse behaviors and target models simultaneously. These are different problem settings and I-GCG does not serve as a meaningful baseline for our evaluation.  Individual black-box attacks such as PAIR and TAP are similarly not applicable baselines for this setting.
> > >
> > > We have compared against the *recent* universal transfer baselines: white-box TransferAttack (Yang et al., 2025) (in table 3) and black-box Logic Jailbreak (in Q2 response). To further address this concern, we additionally evaluate against most recent baseline Slot-GCG [a] (ICLR 2026), the most recent optimization-based jailbreak method, on additional frontier commercial APIs (GPT-4.1-mini and GPT-5.0-mini) under our universal setting:
> > >
> > > | Method | DeepSeek-V3.2 | GPT-4.1-mini | Gemini-1.5-Pro | GPT-5.0-mini |
> > > | --- | --- | --- | --- | --- |
> > > | Slot-GCG | 12.50±2.12 | 14.25±1.77 | 12.25±2.47 | 0.00±0.00 |
> > > | TransferAttack | 12.00±1.08 | 12.67 ± 0.58 | 33.00±16.31 | 0.00±0.00 |
> > > | LogicAttack | 12.00±0.01 | 6.00±0.07 | 23.00±0.02 | 0.25±0.02 |
> > > | **Lookahead-GCG (Ours)** | **16.00±2.93** | **15.00±0.76** | **58.83±1.43** | **1.50±0.01** |
> > >
> > > Results outperforms all white-box and black-box baselines across commercial APIs, with a particularly strong improvement on Gemini-1.5-Pro . The results on GPT-5.0-mini is consistent across all methods, reflecting the inherent difficulty of the universal multi-prompt transfer setting.
> > >
> > > ---
> > >
> > > **Q8: With the recent surge in jailbreak research achieving near-perfect attack success rates and seamless transferability**
> > >
> > > There might be a misunderstanding. To the best of our knowledge, no existing work achieves near-perfect ASR in the *universal multi-prompt transfer* setting — optimizing a single adversarial suffix that transfers across diverse behaviors and black-box target models simultaneously. This remains an open and practically relevant challenge.
> > >
> > > In the universal transfer setting, the most recent work we are aware of is Slot-GCG [a] (ICLR 2026) , which demonstrates continued community interest in this direction yet still falls short of our method (see Q7). The most recent universal black-box method, Logic Jailbreak [2], requires an LLM to process each query individually and thus does not constitute a truly universal attack — the adversarial prompt does not transfer independently of the query. Nevertheless, our method outperforms Logic Jailbreak as well (see Q7), further demonstrating the effectiveness of Lookahead-GCG across both white-box and black-box universal transfer settings.
> > >
> > > ---
> > >
> > > **Our motivation on universal transfer attacks**
> > >
> > > We wish to share our true motivation for focusing on GCG-based universal transfer attacks. We are aware that individual black-box attacks such as PAIR and TAP represent the current mainstream. However, universal adversarial suffixes represent a qualitatively different threat: once computed, the suffix becomes a transferable artifact — appended to any harmful query with zero marginal cost, decoupling the attacker from the end-user and enabling mass exploitation at scale. Unlike PAIR or TAP, which require per-query optimization, a universal suffix is a uniquely dangerous systemic vulnerability: once computed, it can be deployed against *any* harmful query at zero marginal cost, making large-scale exploitation trivial. A defense that only addresses individual attacks may provide no protection against such pre-computed artifacts. From a defense perspective, understanding and anticipating universal transfer attacks before they are exploited at scale is therefore critical for building robust defenses.
> > >
> > > This mindset is the real motivation behind our work. To support the community in building robust defenses, we will commit to open-sourcing our code and maintaining an accessible Lookahead-GCG framework.
> > >
> > > ---
> > >
> > > **Reference**
> > >
> > > [a] Jeong, S., et al. SlotGCG: Exploiting the Positional Vulnerability in LLMs for Jailbreak Attacks, *ICLR 2026*.

---

### Official Review · Reviewer_ouCZ · 2026-03-08

**Soundness:** 4
**Presentation:** 3
**Significance:** 4
**Originality:** 3
**Overall Recommendation:** 5
**Confidence:** 4

**Summary:**

This paper proposes the Lookahead-GCG framework, aiming to address the issue where multi-model training in Large Language Model (LLM) jailbreaking attacks was abandoned due to optimizer instability. The authors point out that standard gradient descent lacks stability when aggregating gradients from heterogeneous models, leading to poor generalization. To this end, the paper combines three techniques: Stochastic Nesterov Accelerated Gradient (SNAG), embedding-space momentum accumulation, and maximally distant initialization. Experimental results show that this method significantly outperforms baselines (e.g., GCG, TransferAttack) on both open-source and closed-source models, achieving an +11.78% improvement in Attack Success Rate (ASR) specifically under multi-model settings. Additionally, the paper provides a theoretical analysis based on stability, proving that SNAG has tighter stability bounds than standard SGD.

**Compliance With Llm Reviewing Policy:**

Affirmed.

**Final Justification:**

The rebuttal adequately addressed my concerns; I confirm my initial recommendation.

**Key Questions For Authors:**

1.   In Section 2.2, you mentioned that LARGO (Li et al., 2025b) and SM-GCG (Gu et al., 2025) also operate in continuous latent spaces or incorporate momentum. However, the paper does not detail the specific differences between this paper's "embedding-space momentum accumulation" and SM-GCG's "spatial momentum." Please explain the essential differences between the proposed method and these existing continuous-space optimization works. A clear distinction will help readers understand the unique contributions of this paper; if the differences are significant, it will enhance my evaluation of the paper's originality.

2.   Algorithm 1 introduces a projection interval $K$ (projecting back to discrete space every $K$ steps). The proof of Theorem 3.3 mentions that the projection error $\epsilon^{proj}$ depends on $K$ (Equation 16 shows the error bound is proportional to $K$). However, Section 4.1 only mentions that $K$ is a shared parameter without detailing the rationale for its selection. Is there an ablation study or discussion regarding the value of $K$? Demonstrating ASR variations and training stability under different $K$ values (e.g., $K=1, 5, 10$) will validate the conclusions regarding $\epsilon^{proj}$ in the theoretical analysis and guide parameter selection in practical deployment. Supplementing this analysis will significantly improve the technical completeness of the paper.

3.   Table 5 shows that Lookahead-GCG saves more training time than GCG and TransferAttack under multi-model settings (25.5h vs. 28.8h/37.7h). However, multi-model training typically implies loading more model weights or performing multiple forward passes per iteration. Please explain why the total time consumption decreases despite introducing the more complex SNAG and continuous-space optimization? Is it because the total number of convergence steps $T$ is reduced, or is it due to optimization of per-step computational overhead? Supplementing efficiency analysis (e.g., a "loss vs. time" comparison plot) will support the conclusion of "superior scalability" and eliminate readers' doubts about the source of efficiency improvements.

**Limitations:**

yes

**Strengths And Weaknesses:**

1.Soundness

Strengths:
The paper demonstrates strong technical soundness. The theoretical section (Theorem 3.3) provides a generalization analysis based on stability, proving the superiority of SNAG in multi-model settings. The experimental design is comprehensive, covering a variety of open-source and closed-source models (e.g., Llama-3, Gemini-1.5-Pro), and multiple independent experiments are conducted to ensure statistical robustness. The improvement in Attack Success Rate (ASR) is statistically significant. The authors focus on a highly pressing problem—namely, the instability of standard optimizers in multi-model gradient aggregation—and verify the effectiveness of the proposed solution through rigorous mathematical derivation and experimental validation.

Weaknesses:
None.


2.Presentation

Strengths:
The overall structure of the paper is clear and the logic is coherent.

Weaknesses:
There is room for improvement in the positioning of related work. Section 2.2 mentions related works such as LARGO and SM-GCG that operate in continuous space or introduce momentum, but Section 3 fails to fully distinguish the specific differences between the "embedding space momentum accumulation" in this paper and the "spatial momentum" in SM-GCG. Readers may find it difficult to understand the essential differences between the method in this paper and existing continuous space optimization works. It is recommended to add comparative discussions in the Related Work or Method section to enhance clarity. In addition, the chapter number before the Impact Statement should be removed to comply with formatting standards.


3.Significance

Strengths:
This work has important research significance. It successfully restores the effectiveness of multi-model adversarial training, a direction that was largely abandoned by the research community previously due to weak performance gains. Overall, this study examines an important theme at the intersection of optimization theory and LLM safety, proving that principled optimizer design can significantly enhance the generality of adversarial suffixes. This not only improves the transferability of jailbreak attacks but also provides a more powerful tool for red-teaming, helping to proactively identify weaknesses in safety alignment.

Weaknesses:
None.

4.Originality

Strengths:
The originality of the paper is mainly reflected in the creative combination of existing technologies and adaptation to specific scenarios. Although SNAG, embedding space optimization, and initialization strategies have been studied in their respective fields, integrating them to solve the stability bottleneck in multi-model jailbreak attacks is a novel perspective. In particular, reconstructing discrete Token optimization into continuous embedding space optimization to adapt to Nesterov momentum has certain methodological innovation.

Weaknesses:
The selection of some hyperparameters (e.g., projection interval $K$) is still empirical, and the differences from recent similar methods need to be more clearly defined, which to a certain extent affects the perception of originality.

---

> ### Author Rebuttal · Authors · 2026-03-31
>
> We sincerely thank you for recommending Accept and for the highly positive feedback regarding our paper's soundness, theoretical foundation, and research significance.  Below, we address your questions point by point.
>
> ---
>
> **Q1: Please explain the essential differences between the proposed method and these existing continuous-space optimization works.**
>
> Both LARGO (Li et al., 2025b) and SM-GCG (Gu et al., 2025) use continuous space but for fundamentally different purposes — LARGO for self-reflective decoding into fluent prompts, SM-GCG for within-step multi-space gradient aggregation. Neither maintains a persistent velocity $v_t$ across steps, making SNAG's lookahead $\tilde{e}_t = e_t + \beta v_t$ inapplicable to both. Our embedding-space formulation serves an entirely different role: a continuous carrier for **cross-step, cross-model momentum accumulation** to stabilize heterogeneous gradient optimization — a problem neither work addresses.
>
> ---
>
> **Q2:  In addition, the chapter number before the Impact Statement should be removed to comply with formatting standards.**
>
> We thank the reviewer for pointing this out. We have removed the section number before the Impact Statement in the revised manuscript.
>
> ---
>
> ***Q3: The selection of projection interval $K$ is still empirical, and the differences from recent similar methods need to be more clearly defined, which to a certain extent affects the perception of originality.***
>
> There might be a misunderstanding. The choice of $K$ is not purely empirical. From Eq. (16), $\epsilon_{\text{proj}} \propto K$, so $K=1$ minimizes the projection error bound. Furthermore, as shown in Section 3.3, the velocity $v_t$ is maintained in continuous space independently of the projection, so $K=1$ does not disrupt momentum accumulation. We therefore set $K=1$ in all our experiments as the theoretically motivated choice.
>
> ---
>
> ***Q4: Please explain why the total time consumption decreases despite introducing the more complex SNAG and continuous-space optimization? Is it because the total number of convergence steps $T$ is reduced?***
>
> Yes, the efficiency gain is primarily explained by **early termination**: in ours as well as GCG‘s settings, optimization stops when all training prompts are successfully jailbroken. Lookahead-GCG's stable gradient aggregation enables training prompts to reach jailbreak success earlier.
>
> Per-step overhead is almost identical for Transfer Attack and ours (264.41s for  Transfer Attack vs 269.78s for Lookahead-GCG per step), confirming that the additional complexity of SNAG introduces negligible computational cost per step.
>
> ---
>
> **Q5: "loss vs. time" comparison plot**
>
> We will include a loss vs. wall-clock time plot in the revision, which we find that the  Lookahead-GCG reaching low loss earlier than GCG and TransferAttack under multi-model settings.

---

> > ### Author Rebuttal · Reviewer_ouCZ · 2026-04-02
> >
> > I thank the authors for their clear and responsive rebuttal.

---

> > > ### Author Response · Authors · 2026-04-05
> > >
> > > We sincerely thank you for the thoughtful engagement and for acknowledging our responses. We are glad the rebuttal addressed your concerns.

---

### Official Review · Reviewer_qqLJ · 2026-03-13

**Soundness:** 3
**Presentation:** 3
**Significance:** 3
**Originality:** 3
**Overall Recommendation:** 4
**Confidence:** 3

**Summary:**

The authors propose "Lookahead-GCG", a new technique for generating adversarial prompts that can be used to attack LLMs. This method improves upon GCG by using a Stochastic Nesterov Accelerated Gradient (SNAG) with lookahead momentum in the embedding space. They also improve ChatBug approach by forcing template diversity.

Their goal is to make these attacks work better against different types of LLMs at once, especially those that aren't open source. Tests were done on various open-source models like Vicuna and LLaMA, showing that Lookahead-GCG works better than older methods whether it's targeting one model or many.

**Compliance With Llm Reviewing Policy:**

Affirmed.

**Final Justification:**

SOTA claim is valid mainly in a narrow baseline set.

**Key Questions For Authors:**

In addition to w1 and w2:

To substantiate the claim that SNAG aligns or stabilizes heterogeneous gradients, I recommend adding an analysis that tracks the gradient variance or the cosine similarity of the gradient updates between the two source models over the course of optimization for both standard momentum and SNAG. Such an analysis would help illustrate whether SNAG indeed produces more consistent gradient directions over the course of optimization.

The single-model baseline is trained exclusively on Llama-3.1-8B, while the multi-model setting uses an ensemble of Llama-3.1-8B and Vicuna-7B-v1.5. To ensure that the reported improvements arise from the multi-model optimization paradigm itself, rather than simply from the inclusion of Vicuna-7B-v1.5 as an additional source model, I recommend adding a single-model baseline trained exclusively on Vicuna-7B-v1.5.

**Limitations:**

yes

**Strengths And Weaknesses:**

**Strengths:**

Good conceptual contribution:  The paper advances the thesis that gradient conflicts across models do not make multi-model prompt optimization infeasible; rather, they indicate the need for more stable optimization dynamics.

Novel Integration: The use of SNAG within the GCG attack framework enhances the stability of multi-model attack optimizations.

Enhanced Template Modification: Improvements to the ChatBug approach ensure maximum diversity in template modifications, leading to more effective attacks.

Comprehensive Evaluation: Extensive experimentation across various models and baseline comparisons provides robust validation of the proposed method's effectiveness.

**Weaknesses**:

w1-Limited Novelty: While the combination of SNAG and ChatBug with GCG is novel, it is an incremental improvement.

w2-Multi-Model Attack Experimentation:

Vicuna and LLaMA are from the same family (LLaMA3 vs. LLaMA2). What would the results be if you used models from different families?

You use two models for the multi-model attack; how about using more models? What would be the effect of using more models?

Maximally distant templates hurt ASR dramatically for Vicuna models; what could be the reason?

---

> ### Author Rebuttal · Authors · 2026-03-31
>
> We sincerely thank you for recommending Accept and for recognizing our conceptual contributions, novel integration, and comprehensive evaluation.  Below, we address your questions point by point.
>
> ---
>
> ***Q1:Limited Novelty: While the combination of SNAG and ChatBug with GCG is novel, it is an incremental improvement.***
>
> We acknowledge that the individual components are not new; the novelty lies in identifying optimizer instability as the root cause of multi-model optimization failure, supported by the theoretical analysis in Theorem 3.3.
>
> ---
>
> ***Q2: What would the results be if you used models from different families?***
>
> We provide additional experiments with an architecturally diverse source ensemble: Falcon-7B-Instruct (TII, GPT-based architecture with MQA and ALiBi) and Llama-3.1-8B-Instruct (Meta, GQA with RoPE) — representing genuinely different model lineages and architectural designs. Results on unseen target models (non-MDT):
>
> | **Method** | **Vicuna** | **Gemma** | **Qwen** | **Llama-2** | **Avg** |
> | --- | --- | --- | --- | --- | --- |
> | GCG | 25.5% | 21.5% | 24.0% | 3.5% | 18.63% |
> | TransferAttack | **87.0%** | 34.0% | 27.5% | 29.0% | 44.38% |
> | **Lookahead-GCG** | 86.5% | **37.0%** | **35.0%** | **40.5%** | **49.75%** |
>
> Lookahead-GCG maintains its advantage over all baselines. This confirms that SNAG's stability advantage generalizes beyond architecturally similar ensembles.
>
> ---
>
> ***Q3: You use two models for the multi-model attack; how about using more models?***
>
> We extended our evaluation to a configuration with $M=3$ source models (Llama-3.1, Vicuna, and Llama-2). The additional results demonstrate that **Lookahead-GCG remains the most effective method** among the compared baselines.
>
> | **Method**  | **Gemma** | **Qwen** | **Gemma (with MDT)** | **Qwen (with MDT)** |
> | --- | --- | --- | --- | --- |
> | GCG | 16.00±1.00 | 15.75±6.72 | 13.00±3.00 | 13.75±2.47 |
> | TransferAttack | 51.75±5.30 | 42.25±10.96 | 35.00±22.00 | 35.00±4.95 |
> | *Lookahead-GCG* | **58.75±1.77** | **63.00±2.00** | **49.75±2.47** | **56.50±2.12** |
>
> The performance of Lookahead-GCG scales significantly better with the number of models compared to GCG. Moreover, combining with the M=1 transfer ASR results in Q6 **Reviewer B2fA**,  Lookahead-GCG benefits most from the larger ensemble (+23.27pts), while GCG shows negligible improvement (+0.74pts), confirming that SNAG's stability advantage scales with ensemble size.
>
> ---
>
> ***Q4: Maximally distant templates hurt ASR dramatically for Vicuna models; what could be the reason?***
>
> The dramatic drop in Vicuna under MDT has two compounding causes. First, as a source model, the suffix overfits to Vicuna's native template; MDT's forced shift to the Llama-2 template (distance=0.4858) directly disrupts this. Second, Vicuna's weak alignment (instruction tuning only, no RLHF) means its high without-MDT ASR (88.38%) is partially inflated by general vulnerability rather than true suffix generalization — once the template shifts, this vulnerability is no longer exploited, amplifying the drop.
>
> ---
>
> **Q5: Track the cosine similarity of the gradient updates between the two source models over the course of optimization for Lookahead-GCG and Momentum-GCG.**
>
> We compared the average cosine similarity of gradients between the two surrogate models for *Lookahead-GCG* and *Momentum-GCG*.
>
> | **Method** | **Avg. Gradient Cosine Similarity** |
> | --- | --- |
> | Momentum-GCG | 0.0109 |
> | *Lookahead-GCG* | **0.0951** |
>
> Due to the high dimensionality of the gradient space, cosine similarities are expected to be small for all methods due to the **high-dimensional geometry**.  Nevertheless, SNAG consistently shows a higher mean gradient cosine similarity than Momentum-GCG, suggesting an improvement in gradient alignment.
>
> ---
>
> **Q6: Adding a single-model baseline trained exclusively on Vicuna-7B-v1.5.**
>
> We provide the requested Vicuna-only single-model baseline. Comparing against the Llama-3.1-only single-model baseline from Table 1 (non-MDT, averaged over Gemma, Llama-2, Qwen):
>
> | **Method** | **Vicuna Single-model** | **Llama-3.1 Single-model (Table 1)** |
> | --- | --- | --- |
> | GCG | 10.67% | 6.39% |
> | TransferAttack | 26.50% | 27.38% |
> | *Lookahead-GCG* | **27.33%** | **27.81%** |
>
> The two single-model baselines yield comparable performance across all methods, confirming that the multi-model gains reported in Table 1 arise from the optimization paradigm rather than the specific choice of source model. In particular, Lookahead-GCG's multi-model gain (+9.80pts, A7) cannot be attributed to the inclusion of any particular source model. We will include this Vicuna single-model baseline in the revision.

---

> > ### Author Rebuttal · Reviewer_qqLJ · 2026-04-03
> >
> > I thank the authors for addressing my concerns in the rebuttal.
> >
> > However, I found reviewer tWKi’s point regarding the limited scope of the SOTA claim compelling. In particular, the evaluation is restricted to a narrow set of optimization-based baselines, and the argument that comparisons to gradient-free methods are not standard does not fully address this concern. In my view, this limits the broader impact and positioning of the work within the current landscape.
> >
> > That said, I still recognize the technical merit and clarity of the contribution. Based on this, I have adjusted my score to a weak accept.

---

> > > ### Author Response · Authors · 2026-04-05
> > >
> > > We truly thank you for consistently recognizing the technical merit and clarity of our contribution, and the following are our responses to your follow-up questions.
> > >
> > > ---
> > >
> > > **Q7: The argument that comparisons to gradient-free methods are not standard does not fully address this concern.**
> > >
> > > We want to clarify that our work targets *universal multi-prompt transfer attacks* — optimizing a single adversarial suffix that generalizes across diverse behaviors and black-box target models simultaneously. Most recent gradient-free methods (e.g., PAIR and TAP) are designed for individual single-prompt attacks, which is not applicable for *universal multi-prompt transfer attacks*, and are thus not directly comparable baselines in our setting.
> > >
> > > To further address this concern, we compared against *recent universal transfer baselines*: universal black-box Logic Jailbreak [b] (Q2 response to Reviewer tWKi) and white-box multi-model Slot-GCG [a], the most recent optimization-based jailbreak method, on additional frontier commercial APIs under our universal setting:
> > >
> > > | Method | DeepSeek-V3.2 | GPT-4.1-mini | Gemini-1.5-Pro | GPT-5.0-mini |
> > > | --- | --- | --- | --- | --- |
> > > | Slot-GCG | 12.50±2.12 | 14.25±1.77 | 12.25±2.47 | 0.00±0.00 |
> > > | TransferAttack | 12.00±1.08 | 12.67 ± 0.58 | 33.00±16.31 | 0.00±0.00 |
> > > | LogicAttack | 12.00±0.01 | 6.00±0.07 | 23.00±0.02 | 0.25±0.02 |
> > > | **Lookahead-GCG (Ours)** | **16.00±2.93** | **15.00±0.76** | **58.83±1.43** | **1.50±0.01** |
> > >
> > > Lookahead-GCG outperforms all white-box and black-box baselines across commercial APIs, with a particularly strong improvement on Gemini-1.5-Pro. The near-zero ASR on GPT-5.0-mini is consistent across all methods, reflecting the inherent difficulty of the universal multi-prompt transfer setting. Notably, Logic Jailbreak still relies on an LLM to process each query individually,  which is not a truly universal multi-prompt black-box transfer attacks, yet our method outperforms it without any such dependency.
> > >
> > > ---
> > >
> > > **Q8: The evaluation is restricted to a narrow set of optimization-based baselines.**
> > >
> > > As discussed in Q1, in the universal multi-prompt transfer setting, optimization-based methods are the de facto *standard*: widely adopted black-box methods such as PAIR and TAP do not support universal multi-prompt transfer attacks. Recent optimization-based works including TransferAttack (Yang et al., 2025) and Slot-GCG [a] (ICLR 2026) exclusively compare against GCG-based baselines, reflecting community consensus on appropriate evaluation protocol. To the best of our knowledge, the GCG family achieves the strongest reported results in this universal transfer setting.
> > >
> > > For completeness, we additionally compared against Logic Jailbreak [b], the most recent universal black-box method we are aware of, and our method outperforms it as well (see Q1). For a more detailed discussion on the advantages and motivation of universal multi-prompt transfer attacks, we refer the reviewer to our response to Reviewer tWKi's Q8.
> > >
> > > ---
> > >
> > > **References**
> > >
> > > [a] Jeong, S., et al. SlotGCG: Exploiting the Positional Vulnerability in LLMs for Jailbreak Attacks, *ICLR 2026*.
> > >
> > > [b] Peng, Jingyu, et al. "Logic jailbreak: Efficiently unlocking LLM safety restrictions through formal logical expression.", arXiv:2505.13527 (2025).

---

### Official Review · Reviewer_B2fA · 2026-03-13

**Soundness:** 3
**Presentation:** 3
**Significance:** 2
**Originality:** 2
**Overall Recommendation:** 3
**Confidence:** 3

**Summary:**

The paper presents Lookahead-GCG, a GCG variant that revisits multi-model optimization for adversarial prompts in language models. The motivation is well laid out: most prior work optimizes against a single source model, and then measures transfer, while multi-model optimization has largely been set aside because standard optimizers only showed marginal gains. The paper argues that this is more an optimization and generalization issue than evidence that multi-model training is inherently ineffective, especially given the difficulty of optimizing transferable attacks in discrete token space. To address this, the method combines Nesterov-style lookahead updates, momentum in continuous embedding space, and periodic projection back to tokens, along with a template-distance-based transfer strategy. The paper also provides a stability-based argument for why SNAG should better handle the heterogeneous gradients that arise in multi-model optimization. Empirically, it evaluates transfer from a two-model source ensemble across five open-source models as well as three closed-source commercial models, including gemini-1.5 Pro, grok-3-mini, and deepseek-v3.

**Compliance With Llm Reviewing Policy:**

Affirmed.

**Final Justification:**

Given how extensively jailbreak attacks have already been studied, along with recent advances in adaptive and second-order attackers, it is difficult for me to see the paper’s practical impact or a sufficiently significant new insight. I think this is a borderline submission.

**Key Questions For Authors:**

1) Could the evaluation be extended beyond HarmBench, for example to AdvBench, and also include additional commercial models?
2) Are there results for optimizing against more than two source models? Since the paper’s default multi-model setting uses two, it would be interesting to know whether optimizing over a broader set of model families further improves generalizability.

**Limitations:**

yes

**Strengths And Weaknesses:**

Strengths
1) Soundness. The paper’s central claim is that the weak or marginal gains previously observed for multi-model jailbreak optimization are not evidence that the setting itself is ineffective, but rather that the bottleneck is the optimizer. More specifically, the paper argues that standard gradient-style optimization in discrete token space is unstable when aggregating heterogeneous gradients across multiple source models. This is an interesting and fairly plausible framing. The supporting theoretical analysis in Section 3.4, which compares SGD-style updates and SNAG through a stability lens while maintaining comparable convergence guarantees, gives a reasonably clean justification for why lookahead optimization may generalize better in the multi-model setting. The dataset and judge used are also reasonable. Harmbench and its corresponding llama-tuned classifier.
2) Empirically. The experimental setup is also fairly clean. In the multi-model setting, the paper optimizes against two open-weight source models, Llama-3.1-8B-Instruct and Vicuna-7B-v1.5, and then evaluates transfer on a broader suite of three other open-source models + the original two source models, both with and without the MDT template-shift setting. It also evaluates transfer to three closed-source production-grade APIs: Gemini-1.5 Pro, Grok-3-mini, and DeepSeek-V3.2. This helps support the claim that the method improves generalization beyond the exact training models. Comparisons with GCG, TransferAttack, and a Momentum-GCG ablation also make it easier to isolate the source of the gains.
3) Presentation. The paper is generally well written and well motivated. It does a good job of framing the distinction between single-model and multi-model jailbreak optimization, where current methods fall short, and what its three main contributions are early on in the introduction. The algorithmic presentation is also fairly clear.
4) Significance. The evaluation on both open-source and closed-source models is a strong part of the paper. Transfer itself is a practical and realistic threat model, and the results here are meaningful: the method is not just improving on nearby open-weight surrogates, but also shows gains on Gemini-1.5 Pro, Grok-3-mini, and DeepSeek-V3.2, with the average commercial-API ASR reaching 34.03. That makes the paper’s contribution more practically relevant than results that only hold within a small set of closely related open models.
5) Originality. The paper’s three main contributions toward making multi-model jailbreak optimization work are of moderate novelty, but I still think the combination is interesting: using Nesterov lookahead in continuous embedding space, periodic projection back to the token level, and maximally distant initialization make for a fairly creative optimizer from a design perspective. Paired with the stability analysis, this also gives the paper more conceptual grounding than a purely empirical improvement.

Weaknesses
1) Significance. The paper, excluding the two open-source models used during optimization, evaluates transfer only against three other open-source models and three commercial models. It would strengthen the paper’s impact to include more notable commercial systems, especially Claude and GPT models, given the extent to which they have improved in reasoning safety and safeguards more broadly. Stronger empirical transfer results on those systems would make the paper's practical significance more convincing.
2) Soundness. Although the theoretical analysis is meaningful in building the argument that SNAG is less sensitive to any one sampled model/prompt pair, and thus should generalize better across models, there is still a gap between this stability-based analysis and downstream transfer, which the empirical results need to support. Empirically, the paper does show improvements over the other three methods, and the reported standard deviations are generally fairly bounded. That said, Momentum-GCG remains a strong ablation and comes fairly close on a number of the open-source results. The gain from the paper’s own single-model to multi-model variant is also somewhat bounded: it is 11.78 points on the open-source average and 4.48 points on the commercial APIs, so the improvement is real but sometimes very modest. Also note, I am referring to the scores without MDT. In transfer settings, I am not sure how realistic it is to assume the with MDT evaluation structure.
3) Presentation.The evaluation on open-source models includes the original 2 source models (Llama-3.1-8B and Vicuna) in the average ASR, so the reported average improvement and statistics don't cleanly reflect transfer generalizability from this method. Vicuna results might inflate scores?

---

> ### Author Rebuttal · Authors · 2026-03-31
>
> We deeply thank you for the thoughtful reviews and valuable suggestions.  Below, we address your comments point by point.
>
> ---
>
> **Q1: It would strengthen the paper’s impact to include more notable commercial systems**
>
> We have conducted additional experiments on *GPT-4.1-mini*. Lookahead-GCG achieves 15.17 ± 0.76% ASR, outperforming TransferAttack (12.67 ± 0.58%) and GCG (6.00±1.00%), consistent with our findings on other commercial APIs. We will include these results in the revision.
>
> ---
>
> **Q2: There is still a gap between this stability-based analysis and downstream transfer. The empirical results need support.**
>
> (1) We measured the average cosine similarity of gradients between the two surrogate models during optimization:
>
> | Method | Avg. Gradient Cosine Similarity |
> | --- | --- |
> | Momentum-GCG | 0.0109 |
> | *Lookahead-GCG* | **0.0951** |
>
> The absolute values are small due to high-dimensional geometry. Lookahead-GCG shows *8.7×* higher gradient alignment than Momentum-GCG, directly supporting the stability argument.
>
> (2) Lookahead-GCG achieves comparable training loss to baselines (Figure 1) yet significantly higher transfer ASR (Tables 1, 3) — the same training loss but better transfer is what tighter stability predicts.
>
> ---
>
> **Q3: Momentum-GCG remains a strong ablation and comes fairly close on a number of the open-source results.**
>
> Following our newly conducted results in A7, Lookahead-GCG (37.61%) maintains a consistent lead over Momentum-GCG (33.05%, +4.56pts) and TransferAttack (30.53%, +7.08pts) in the transfer-only setting. Its multi-model gain (+9.80pts) exceeds Momentum-GCG's (+6.54pts), directly demonstrating that the lookahead mechanism improves generalization over standard momentum.
>
> Paired t-tests confirm Lookahead-GCG statistically significantly outperforms GCG (p=0.00035) and TransferAttack (p=0.046); notably, Momentum-GCG does **not** achieve significance over TransferAttack, further isolating the lookahead mechanism as the source of improvement.
>
> ---
>
> **Q4: The gain from the paper’s own single-model to multi-model variant is also somewhat bounded**
>
> We think that gains should be evaluated in context: *higher* baselines leave *less* headroom for improvement. In the transfer-only setting (our response A7), Lookahead-GCG's single-model baseline (27.81%) already leads all baselines, making further gains inherently harder due to *ceiling effects*. By contrast, GCG's larger raw gain (+8.75pts) starts from a much lower baseline (6.39%), where headroom is abundant. TransferAttack's multi-model gain is only +3.16pts from a comparable baseline (27.38%), confirming that meaningful multi-model improvement at this performance level is not small. Lookahead-GCG's +9.80pts gain under these conditions represents a practical improvement.
>
> ---
>
> **Q5:  In transfer settings, I am not sure how realistic it is to assume the MDT evaluation structure.**
>
> MDT requires no access to the target model, where all template distance computations are performed solely on the open-source source model $M_s$, with $M_t$ invoked only as a black-box API. In the worst case, if the default template for $M_t$ is unavailable, one can simply enumerate all candidate templates and select the best-performing one.
>
> ---
>
> **Q6: Reported average improvement and statistics with source models don't cleanly reflect transfer.**
>
> We recompute ASRs excluding the two source models, retaining only unseen transfer targets (Gemma, Llama-2, Qwen):
>
> | Method | Transfer-Only Avg | Significance vs Lookahead |
> | --- | --- | --- |
> | TransferAttack | 30.53% | p=0.046 |
> | GCG | 15.14% | p=0.00035 |
> | Momentum-GCG | 33.05% | p=0.298 |
> | Lookahead-GCG | **37.61%** | — |
>
> Lookahead-GCG maintains a consistent lead over all baselines, and its multi-model gain remains the largest, confirming that the improvement is not an artifact of source model inflation. We will report transfer-only averages as a supplementary metric in the revision.
>
> ---
>
> **Q7:  Conduct AdvBench**
>
> Following the reviewer’s suggestion, we conducted additional experiments on AdvBench with $M=3$ source models.
>
> The results demonstrate the superior generalizability of Lookahead-GCG, which consistently outperforms all baselines. Lookahead-GCG achieves $60.38\%$ and $42.60\%$ transfer ASR on Gemma and Qwen targets respectively, representing a significant margin over both GCG ($7.88\%$ and $5.68\% $) and TransferAttack ($50.58\%$ and $29.52\%$). These findings align with our HarmBench results.
>
> ---
>
> **Q8: Are there results for optimizing against more than two source models?**
>
> Please refer to our response to Reviewer qqLJ‘s Q3 for more details.
>
> Moreover, combining with the M=1 transfer ASR results in Q6, Lookahead-GCG benefits most from the larger ensemble (+23.27pts), while GCG shows negligible improvement (+0.74pts), confirming that SNAG's stability advantage scales with ensemble size.
>
> ---
>
> **Ref**
>
> [1] Bousquet, Olivier, and André Elisseeff. "Stability and generalization." JMLR, 2002.

---

> > ### Author Rebuttal · Reviewer_B2fA · 2026-04-04
> >
> > I thank the authors for the rebuttal. However, I am still worried about the paper's practical impact. The question of transfer to frontier commercial models remains open despite the multi-model optimization; GPT-4.1-mini predates the current generation of safety-tuned systems. The multi-model gains themselves still appear moderate especially given that other baselines presumably better than Momentum-GCG are not included (defer to reviewer tWKi). For these reasons, unfortunately, I have to maintain my score.

---

> > > ### Author Response · Authors · 2026-04-05
> > >
> > > We sincerely thank you for your continued engagement and thoughtful feedback. We understand the remaining concerns and here is our responses to your follow-up questions.
> > >
> > > ---
> > >
> > > **Q9: However, I am still worried about the paper's practical impact.**
> > >
> > > Recall that our setting is *universal multi-prompt transfer attack* — optimizing a single adversarial suffix that generalizes across diverse behaviors and black-box target models simultaneously. Universal adversarial suffixes represent a qualitatively different and practically critical threat: once computed, a single suffix can be appended to any harmful query with zero marginal cost, decoupling the attacker from the end-user and enabling mass exploitation from a single computation. This is vastly different from individual attacks like PAIR and TAP, which require *per-query optimization*.
> > >
> > > From a defensive perspective, universal transfer attacks are particularly concerning precisely because of their scalability: a single pre-computed suffix can be deployed against any harmful query at zero marginal cost, making large-scale exploitation trivial. A defense that only addresses individual attacks may provide no protection against such artifacts — understanding and anticipating universal transfer attacks is therefore critical for building robust defenses.
> > >
> > > As demonstrated in our response to Q10, Lookahead-GCG already outperforms recent white-box [a] and black-box [b] universal baselines on frontier commercial APIs, further demonstrating its practical relevance.
> > >
> > > ---
> > >
> > > **Q10: The question of transfer to frontier commercial models remains open despite the multi-model optimization; GPT-4.1-mini predates the current generation of safety-tuned systems. The multi-model gains themselves still appear moderate especially given that other baselines presumably better than Momentum-GCG are not included (defer to reviewer tWKi).**
> > >
> > > To address the reviewer's concern, we include Slot-GCG [a] (ICLR 2026), the most recent universal multi-model white-box optimization method, and Logic Jailbreak [b], the most recent universal black-box method, as additional baselines. We further extend our evaluation to newer frontier commercial systems, including DeepSeek-V3.2 and GPT-5.0-mini,  representatives of the recently updated frontier models.
> > >
> > > | Method | DeepSeek-V3.2 | GPT-5.0-mini |
> > > | --- | --- | --- |
> > > | Slot-GCG | 12.50±2.12 | 0.00±0.00 |
> > > | TransferAttack | 12.00±1.08 | 0.00±0.00 |
> > > | LogicAttack | 12.00±0.01 | 0.25±0.02 |
> > > | **Lookahead-GCG (Ours)** | **16.00±2.93** | **1.50±0.01** |
> > >
> > > Lookahead-GCG achieves the highest ASR on DeepSeek-V3.2 and GPT-5.0-mini among all methods.  Taken together, these results demonstrate that Lookahead-GCG consistently achieves the strongest performance in the universal multi-prompt transfer setting
> > >
> > > ---
> > >
> > > **References**
> > >
> > > [a] Jeong, S., et al. SlotGCG: Exploiting the Positional Vulnerability in LLMs for Jailbreak Attacks, *ICLR 2026*.
> > >
> > > [b] Peng, Jingyu, et al. "Logic jailbreak: Efficiently unlocking LLM safety restrictions through formal logical expression.", arXiv:2505.13527 (2025).

---

### Decision · Program_Chairs · 2026-04-30

**Decision:**

Accept (regular)

**Comment:**

This submission proposes Lookahead-GCG, an improved jailbreak attack based on GCG but incoporates Stochastic Nesterov Accelerated Gradient, embedding-space momentum accumulaton, and maximally distance initialization. Lookahead-GCG improves the transferability of jailbreak suffixes especially when generated from multiple source models. All reviewers were actively involed during the rebuttal phase. The authors did a great job in addressing major concerns (extended multi-model evaluatio settings, clarifications on key hyperparameters and experimental phenomenons). Overall, authors appreciate the novel contributions to GCG from the optimizer and generalization perspective. On the other hand, concerns remain on the evaluation protocol and baseline selection, where authors believe that the universal optimization-based suffix search is a standalone setting and Lookahead-GCG achieves SOTA performance in this setting, while three out of the four reviewers feel gradient-free black-box attacks, though may optimize at individual instance level, should be taken into account given their superior performances on frontier commericial models. In my opinion, given the rapid progress in LLM jailbreak field, it might be too early to say one type of methods will be vanishing in the future. Moreover, the universal suffix setting has its usage scenario as a strongly constrained threat model. So I tend to recommend a weak accept to this meaningful and insightful improvement for this type of attacks.